# Thermodielectric Properties of Polyurethane Composites with Aluminium Nitride and Wurtzite Boron Nitride Microfillers: Analysis Below and near Percolation Threshold

**DOI:** 10.3390/s25134055

**Published:** 2025-06-29

**Authors:** Alexey Gunya, Jozef Kúdelčík, Štefan Hardoň, Marián Janek

**Affiliations:** 1Department of Physics, Faculty of Electrical Engineering and Information Technology, University of Zilina, Univerzitná 8215/1, 010 26 Žilina, Slovakia; gunya@fyzika.uniza.sk (A.G.); stefan.hardon@uniza.sk (Š.H.); marian.janek@uniza.sk (M.J.); 2Department of Astronomy and Astrophysics, Universität Potsdam, Karl-Liebknecht-Straße 24/25, 14476 Potsdam, Germany

**Keywords:** composites, polyurethane, filler, thermal conductivity, dielectric permittivity

## Abstract

This study explores microcomposites’ thermodielectric properties—thermal conductivity ($k_{\mathrm{eff}}$) and dielectric permittivity ($\varepsilon_{r}$)—across filler concentrations from 1 wt% (φ≈0.0035) to 60 wt% (φ≈0.45) spanning the pre- (φ<0.16) and within-percolation threshold (0.16≤φ≤0.29). Thermal measurements were conducted using a newly designed, cost-effective thermal measurement setup. The setup utilised a transient heat pulse methodology with a heater and NTC thermistors, with a precision better than ±0.01 W·$m^{-1}\cdot$$K^{-1}$. Dielectric properties were measured using a three-electrode system over a broad frequency and temperature range. The measurements demonstrate an effective thermal conductivity $k_{\mathrm{eff}}$ of 0.72 W·$m^{-1}\cdot$$K^{-1}$ for AlN at φ=0.36 and 0.65 W·$m^{-1}\cdot$$K^{-1}$ for wBN already at φ=0.12. Although theoretical models suggest that, considering interfacial Kapitza resistance, it can yield a $k_{\mathrm{eff}}$ corresponding to approximately 1–3% of the conductivity of pure material filler, the experimental measurements indicate a maximum of around 0.5%. Dielectric measurements show that in comparison to pure polyurethane, the presence of 60% AlN or 40% wBN at 60 °C decreased the loss tangent by 20 times in the condition of a quasistatic electric field.

## 1. Introduction

Polymer-based composites, particularly those incorporating functional micro- and nanoscale particles, have garnered significant attention in recent years due to their ability to enhance the thermal, mechanical, and dielectric properties of polymers beyond those of the neat matrix [1,2]. Polyurethane (PUR), a segmented copolymer composed of soft polyol segments and rigid isocyanate-based hard segments, is widely used across multiple industries, from coatings and structural foams to electronic encapsulants and thermal interface materials [3]. Its excellent processability, low intrinsic thermal conductivity ($k_{\mathrm{PUR}}=0.2$ W·$m^{-1}\cdot$$K^{-1}$), and high dielectric strength make it an ideal candidate for multifunctional composite development [4].

The PUR’s versatility arises from its phase-separated morphology and tunable segmental composition, enabling the optimisation of mechanical, thermal, and dielectric properties through the incorporation of nanofillers [5,6,7]. Previous studies have confirmed that the addition of ceramic particles (e.g., ${\mathrm{Al}}_{2}$$O_{3}$, ZnO, and MgO) significantly affects both polymer chain mobility and the dielectric relaxation mechanisms, as evidenced by shifts in the glass transition temperature and low-frequency permittivity behaviour. Such insights underscore the importance of tailoring filler–matrix interactions and dispersion quality to achieve targeted performance.

The synergistic combination of a flexible polymer and high-performance fillers leads to multifunctional benefits that surpass the properties of each constituent individually [8]. In particular, composites incorporating ceramic fillers—such as aluminium nitride (AlN) and wurtzite-phase boron nitride (wBN)—have attracted significant interest for thermal management and dielectric applications [9,10,11]. Polymer composites reinforced with AlN and wBN particles exhibit strong potential for industrial applications, particularly in the battery sector, due to their favourable thermodielectric properties. With an average particle diameter of 5–10 µm, the composite in the work is categorised as a microcomposite. Such a particle scale of a filler offers advantages in mechanical robustness and processability at high loadings [12,13]. This study focuses on evaluating the dielectric performance of the composites under both DC and AC conditions, where distinct polarisation and conduction mechanisms govern energy dissipation and insulation behaviour.

The rapid advancement of high-performance electronics, electric aircraft, and energy storage systems has driven a surge in demand for advanced thermal interface materials and dielectric composites [14]. The incorporation of thermally conductive microparticles into polymers significantly enhances heat transport by forming efficient conduction pathways within the matrix. AlN is recognised for its high thermal conductivity (k≈250 W·$m^{-1}\cdot$$K^{-1}$ [15]) and moderate dielectric constant, making it highly effective for enhancing heat dissipation while ensuring electrical insulation [16]. Similarly, wBN (k≈700 W·$m^{-1}\cdot$$K^{-1}$ [17]) is emerging as a cost-effective alternative to graphene fillers due to its excellent thermal stability, low dielectric loss, and high breakdown strength [18,19].

In the DC regime, characterised by quasistatic electric fields, the composite material must maintain strong insulating properties to suppress leakage currents and ensure operational safety, particularly in battery systems. Conversely, under AC conditions relevant to modular battery systems equipped with DC/AC converters, the material should demonstrate low dielectric losses, specifically minimal energy dissipation resulting from permittivity and loss tangent, when subjected to time-varying electric fields. This contributes to reduced inefficiencies and improved thermal stability.

One of the key parameters influencing the performance of composite materials is the percolation threshold, which represents the critical filler concentration at which a continuous conductive or structural network forms within the polymer matrix. Below this threshold, the thermal and dielectric properties are predominantly influenced by the individual contributions of the fillers and the interfacial interactions. However, as the filler content approaches or exceeds the percolation threshold, significant enhancements in thermal conductivity and dielectric permittivity occur as a result of improved phonon transport and increased polarisation effects [20].

In this study, PUR-AlN and PUR-wBN microcomposites were studied by analysing the thermal conductivity and dielectric permittivity across filler concentrations ranging from 1 wt% (φ≈0.0035) to 60 wt% (φ≈0.45), thereby covering regions both below and near the percolation threshold ($\varphi_{c}\approx$ 0.16–0.29). This investigation addresses the following:The variation in thermal conductivity *k* and dielectric permittivity ε with respect to filler type, volume fraction 0.035<φ<0.4, and temperature 25 °C < 120 °C.The transition from isolated filler contributions for volume fraction φ<0.16 to the establishment of a connected filler network and its impact on the overall properties for volume fractions 0.16<φ<0.29 and higher.Comparison of the experimental results with the theoretical model proposed by Nan et al. [21].The addressal of the fabrication challenges at high filler loadings and proposal of scalable processing routes.An outline of the prospects for hybrid filler systems and practical application in next-generation battery modules for electric aircraft.

To perform thermal conductivity measurements, the appropriate facility must be used. In this study, specifically, a dedicated setup was designed. It is a relatively accurate (within 5%) and cost-efficient system suitable for corresponded research, addressing limitations associated with some existing thermal analysis techniques such as the Angstrom method [22,23], steady-state methods like the guarded hot plate [24], laser flash analysis [25], or the transient plane source method [26], which can involve complex setups, specific sample requirements, or higher costs. This setup is described in detail in Section 3.3.

This study explores the thermal and dielectric properties of composites incorporating microparticles of AlN and wBN, focusing on their behaviour in the pre-percolation and efficient percolation regimes. The observed trends exhibit a strong correlation with analytical models of thermal conductivity as a function of filler volume fraction, providing critical insights into the material’s performance. These findings lay a solid foundation for the development of large-scale industrial applications, particularly for high-performance thermally conductive materials in next-generation battery systems. Furthermore, the results offer a reliable platform for future experiments exploring the hybridisation of AlN and wBN fillers to investigate potential synergistic enhancements.

## 2. Theory

The theoretical framework considers the interactions between the polymer matrix and the filler particles and their collective impact on the composite’s overall properties. This section details theoretical models that describe the behaviour of PUR-AlN and PUR-wBN microcomposites, covering thermal conductivity, dielectric permittivity, and percolation theory alongside various predictive frameworks.

### 2.1. Thermal Conductivity

Thermal conductivity (*k*) measures a material’s ability to conduct heat and is described by Fourier’s Law:(1)q=−k∇T,
where *q* is the heat flux (W/$m^{2}$), and ∇T denotes the temperature gradient (K/m). The heat is primarily transferred through phonons—quantised lattice vibrations. PUR, with its amorphous structure, causes significant phonon scattering, resulting in a low thermal conductivity, similar to other thermal insulators like polystyrene. In contrast, the cubic lattice of AlN facilitates efficient phonon transport, and BN’s wurtzite structure yields an even higher thermal conductivity, approaching that of copper while remaining electrically insulating.

In composite materials, the overall or effective thermal conductivity ($k_{\mathrm{eff}}$) is influenced by the heterogeneous nature of the composite, wherein heat alternates between transmission through the polymer matrix and the thermally conductive filler. Several factors affect the $k_{\mathrm{eff}}$, which we present below:Filler Conductivity ($k_{f}$): A higher $k_{f}$ improves $k_{\mathrm{eff}}$; however, the low *k* of the polymer matrix tempers the enhancement.Volume Fraction (φ): An increasing filler volume fraction creates additional conductive pathways, although the benefit may level off due to interfacial limitations.Interfacial Thermal Resistance: This resistance, commonly known as the Kapitza resistance ($R_{k}$), arises from mismatches in the phonon properties between the soft organic bonds in PUR (e.g., C–H and C–O) and the rigid bonds in AlN (Al–N) or wBN (B–N). Typical values of $R_{k}\approx 10^{-7}\;m^{2}K/W$ yield an interfacial conductance of $G_{k}=1/R_{k}\approx 10^{7}\;{W/m}^{2}K$.Filler Geometry: Spherical fillers (aspect ratio ρ=1) are generally less effective than anisotropic fillers, such as platelets or rods (ρ=5−10), which can align to create more efficient thermally conductive chains.

Interfacial resistance plays a crucial role in determining the effective thermal conductivity of composites. For instance, a PUR-AlN composite for purely spherical particles of filler with a sufficient volume fraction corresponded to efficient percolation φ=0.3 might ideally (without resistance) reach an effective thermal conductivity of $k_{\mathrm{eff}}=75$ W·$m^{-1}\cdot$$K^{-1}$ (which is 30% of $k_{AlN}∼250$ W·$m^{-1}\cdot$$K^{-1}$). However, due to interfacial resistance, the practical $k_{\mathrm{eff}}$ may be limited to approximately 0.5–2 W·$m^{-1}\cdot$$K^{-1}$. This discrepancy underscores the challenge posed by phonon scattering at the polymer–filler interface. Moreover, the phonon mean free path (λ) in AlN is around 50 nm but diminishes to 5–10 nm in the PUR interface. Techniques such as surface functionalisation (e.g., silane coupling agents) can potentially enhance $G_{k}$ to $∼10^{7}\;W\cdot m^{-2}K^{-1}$, which in turn may significantly improve $k_{\mathrm{eff}}$ [27].

### 2.2. Dielectric Permittivity

Dielectric permittivity quantifies a material’s capability to store electrical energy when exposed to an external electric field. It represents how much the electric field is reduced within the medium compared to a vacuum. In composites, dielectric permittivity is crucial for evaluating the insulating performance under both AC and DC conditions as well as varying temperatures.

For alternating current (AC) fields, the complex relative permittivity $\varepsilon^{*}\left(\omega \right)$ is given by(2)$$\varepsilon^{*}\left(\omega \right)=\varepsilon_{r}\left(\omega \right)-j\varepsilon_{i}\left(\omega \right),$$
where $\varepsilon_{r}\left(\omega \right)$ is the real part (the relative dielectric constant) representing the material’s ability to store energy, and $\varepsilon_{i}\left(\omega \right)$ is the imaginary part representing the dielectric losses. The loss factor, or dissipation factor, is defined as(3)$$\tan\delta =\frac{\varepsilon_{i}\left(\omega \right)}{\varepsilon_{r}\left(\omega \right)}.$$

In nonpolar dielectrics, the permittivity is nearly frequency-independent, whereas weakly polar dielectrics follow the Debye model [28] with one characteristic relaxation time.

In realistic dielectric materials, conductive losses—resulting from the movement of free charges—also contribute to the overall energy dissipation, especially at low frequencies. The conductive loss is described by Jonscher’s Law [29] as(4)$$\varepsilon_{i}=\frac{\sigma }{\varepsilon_{0}\omega^{n}},$$
with n∈(0,1) being a dielectric exponent and σ being the DC conductivity.

For materials with a distribution of dipolar orientations and relaxation times, the Cole–Cole model refines the description of dielectric relaxation. Other empirical models, such as Cole–Davidson, can also be used to fit experimental data from dielectric relaxation spectroscopy [30]. In this work, the Havriliak–Negami model is applied, which is defined as(5)$$\varepsilon^{*}=\varepsilon_{\infty }+\frac{\Delta \varepsilon }{{\left(1+{\left(j\omega \tau \right)}^{\left(1-\alpha \right)}\right)}^{\beta }}-j\frac{\sigma }{\varepsilon_{0}\omega },$$
where $\varepsilon_{\infty }$ denotes the high-frequency limit of permittivity reflecting, while Δε quantifies the total dielectric strength related to the number and effective dipole moment of participating dipoles. It is the difference between the static (low-frequency) and the high-frequency limit of permittivity, i.e., $\Delta \varepsilon =\varepsilon_{s}-\varepsilon_{\infty }$. τ represents the characteristic relaxation time, defining the average speed of dipole reorientation. Furthermore, α,(0<α≤1) is the shape parameter for asymmetry, where values higher than zero indicate a skewed distribution of relaxation times on the high-frequency side, suggesting cooperative molecular motions. The next parameter β,(0<β≤1) is the shape parameter for broadness, with values less than one signifying a broader distribution of relaxation times, often attributed to material heterogeneity. When α=0,β=1, the model reduces to the classical Debye behaviour [28].

For composite systems, the effective permittivity $\varepsilon_{\mathrm{eff}}$ can be estimated through theoretical models like the Bruggeman model:(6)$$\varepsilon_{\mathrm{eff}}=\varepsilon_{\mathrm{matrix}}\left(\frac{\sum_{i}\varepsilon_{i}\varphi_{i}}{\sum_{i}\varphi_{i}}\right),$$
where $\varepsilon_{\mathrm{matrix}}$ is the dielectric constant of the polymer matrix, and $\varepsilon_{i}$ and $\varphi_{i}$ are the permittivity and volume fraction of the individual filler components.

### 2.3. Percolation

Enhancing the thermal conductivity of a polymer matrix through the addition of fillers is one of the most effective methods without altering the polymer’s chemical composition. The efficiency of this approach depends on both the intrinsic thermal conductivity of the filler and its concentration within the matrix. At low concentrations, fillers—often approximated as spherical particles—are isolated within the polymer. As the concentration increases, the probability of direct contact between fillers grows, eventually leading to the formation of interconnected particle chains. These chains establish continuous thermal pathways, significantly enhancing heat conduction through the composite.

This phenomenon, known as percolation, represents a critical transition in composite systems. Below the percolation threshold, the thermal and dielectric properties are dominated by the isolated filler–matrix interactions. Once the concentration exceeds the critical percolation threshold, a continuous network forms, leading to rapid changes in both thermal conductivity and dielectric permittivity. The percolation phenomenon in composites is often characterised by a power law behaviour:(7)$$k_{\mathrm{eff}}∼{\left(\varphi -\varphi_{c}\right)}^{t},$$
where *t* is a critical exponent indicative of the connectivity in the system. The percolation threshold is typically estimated to be in the range of 0.16 [31] to 0.29 [32], depending on factors such as particle shape and distribution. Beyond $\varphi_{c}$, the composite’s thermal conductivity increases markedly before gradually saturating as the filler concentration nears 100%.

### 2.4. Thermal Conductivity and Dielectric Permittivity Prediction Framework

Traditional experimental approaches to developing composite materials are time-consuming and resource-intensive. In contrast, analytical and numerical simulations can efficiently determine the key parameters that govern composite behaviour, fostering a more targeted and scientifically informed fabrication process.

Accurate predictions of the effective thermal conductivity $k_{\mathrm{eff}}$ in composite materials require models that account for the filler’s characteristics, its distribution, and the interactions with the matrix. For composites with spherical fillers and low filler content, the rule of mixtures provides a simple estimate:(8)$$k_{\mathrm{eff}}=\varphi \;k_{\mathrm{filler}}+\left(1-\varphi \right)\;k_{\mathrm{matrix}},$$
where φ is the volume fraction of the filler, $k_{\mathrm{filler}}$ represents the filler’s thermal conductivity, and $k_{\mathrm{matrix}}$ is that of the polymer matrix.

However, this linear approximation becomes inadequate at higher filler loadings when the fillers begin to interact or form networks. For such cases, models such as, for example, the Maxwell–Eucken model or those similar offer a more refined prediction by incorporating filler geometry and interfacial effects. Several theoretical frameworks have been developed to predict the effective properties of composites across various filler concentrations via Effective Medium Theory [33], Wiener Bounds [34], Agrawal’s Model [35], Ngo and Byon Approximations [36], Hashin–Shtrikman Bounds [37], and Nan Bounds [21].

The Nan model stands out as the most promising among the listed models due to its versatility and precision in predicting the effective thermal conductivity of composite materials. Developed by Nan et al. in 1997, it accounts for particle shape, size, and interfacial thermal resistance, which are critical for modern nanomaterials. Unlike the Effective Medium Theory or Wiener Bounds, which assume idealised conditions, Nan’s model incorporates realistic microstructural features, making it more applicable to complex systems. Compared to Agrawal’s Model or Ngo and Byon Approximations, it offers a broader range of applicability across different material compositions. The Hashin–Shtrikman Bounds provide theoretical limits but lack the predictive specificity of Nan’s approach. Its ability to handle nanoscale effects and align with experimental data positions the Nan model as a superior tool for advancing thermal management in nanotechnology.

In this work, while a predictive model for dielectric permittivity is acknowledged, the focus is on the Nan model for thermal conductivity prediction, where in the approach for monocomponent composites, these bounds additionally factor in contact resistivity and the geometric attributes of the filler.

## 3. Materials and Methods

### 3.1. Materials

In this work, for fabrication of composite, the PUR VUKOL N22 MAGNA BLUE manufactured by the company VUKI a.s. (Bratislava, Slovakia) [38] was used. This is a two-component PUR compound for potting, delivered separately as a combination of polyol resin and hardener with low initial viscosity. After mixturing, this compound hardens at room temperature, making it suitable for a wide range of applications. One of the main advantages of this product is that it contains only small amounts of inorganic fillers and is free of solvents (VOC < 1%). This type of PUR is intended for potting and casting of electric batteries and is used a wide range of applications. PUR was mixed in a ratio of 100:47 with VUKIT M hardener, with a hardening time of approximately 10 min and a full hardening time of 7 days. Basic parameters include the following: Thermal class B (130 °C); dielectric strength: 20–25 kV/mm, flame retardancy: V2; permittivity: 4.2; volume resistivity: $6.5\times 10^{11}$
Ω·cm; $T_{g}$: 15.6 °C. The set of prepared compositions is summarised in Table 1, where the filler contents are initially expressed in weight percent (wt%). For graphical analysis, these values were converted to volume fractions (φ) using the following equation:(9)$$\varphi =\frac{\frac{wt_{i}}{\rho_{i}}}{\sum_{j}\frac{wt_{j}}{\rho_{j}}},$$ Only the binary filler–matrix formulations AlN + PUR and wBN + PUR were considered, where densities were $\rho_{\mathrm{AlN}}=3.3\;{g/\mathrm{cm}}^{3}$, $\rho_{\mathrm{wBN}}=2.1\;{g/\mathrm{cm}}^{3}$, $\rho_{\mathrm{PUR}}=1.2\;{g/\mathrm{cm}}^{3}$. Microcomposite samples were synthesised in a controlled laboratory environment using a direct dispersion method, as described in previous works [39,40,41].

### 3.2. Fabrication

For sample preparation (Figure 1), a dispersive mixing protocol was implemented to minimise particle agglomeration within the liquid polymer matrix, thereby ensuring a uniform distribution of fillers throughout the volume. Prior to incorporation into the two-component (2C) system, the particles were subjected to vacuum drying at 40 °C for 24 h under ambient laboratory pressure to eliminate surface-adsorbed moisture. To achieve an optimal viscosity for mixing, the polyol—serving as the first component of the 2C system—was preheated to 40 °C. The dried particles were then accurately weighed to reach the desired filler concentrations (up to 60 wt%) and subsequently introduced into the polyol phase.

The dispersion process consisted of mechanical mixing using an electric stirrer at 500 rpm for 3 h at 40 °C. An electric stirrer was chosen instead of a magnetic stirrer due to the increased viscosity and elevated filler content (above 10 wt%), which necessitated higher shear forces to ensure effective particle dispersion within the polymer matrix. This step was followed by vacuum treatment and subsequent ultrasonic homogenization using a probe sonicator for 1 h to further enhance particle dispersion and minimise the formation of agglomerates. A final degassing procedure was carried out at 10 mbar for 1 h under continuous stirring to eliminate entrapped air from the suspension.

Due to the high surface energy and cohesive nature of the particles employed in thermally and dielectrically functional composites, proper dispersion is critical. Insufficient distribution can adversely affect the thermal, dielectric, and mechanical properties of the resulting material and may also influence space charge accumulation phenomena. The application of ultrasonic agitation proved particularly effective in disintegrating particle aggregates and promoting uniform distribution.

Subsequently, the hardener (second component of the 2C system) was added to the mixture at the recommended mass ratio of 100:47 (polyol to hardener). This protocol ensured homogeneous dispersion of particles within the polymer matrix, even at high loading levels. The final suspension was then cast into disk-shaped moulds (80 mm × 3 mm) and rectangular moulds (25 mm × 40 mm × 3 mm) for curing and further characterisation.

The result of fabrication can be seen in Figure 2. The image of the fabricated PUR + 20% AlN sample, made with SEM, is presented here. Here, the Phenom G6 Desktop SEM (Thermo Fisher SCIENTIFIC, Slovakia) was used. It is equipped with a backscattered electron detector to verify particle dispersion. Observations were made under low vacuum and in charge reduction mode, with an applied acceleration voltage of 10 kV. The surface analysis indicates an almost homogeneous distribution of AlN particles with separate contact chains between them, which is natural for concentrations near the percolation level.

### 3.3. Thermal Conductivity Measurements

The developed setup utilises a transient heat pulse methodology [42]. A plane heating element acts as the heat source, positioned between two identical cylindrical samples of the material being investigated. These samples feature specific grooves to accommodate the heater and two NTC thermistors [43]. One thermistor is placed between the heater and the first sample to capture the initial temperature rise, while the second is located on the opposite face of the first sample (between the first and second samples) to detect the heat pulse arrival. A third identical sample is placed atop the second to improve heat flow uniformity and minimise boundary effects. This assembly is maintained under consistent pressure using a 3D-printed gripper with a torque adapter situated between passive coolers to ensure a stable starting temperature. Custom electronics were integral to the design for precision and affordability. A low-current source utilising a TL431 regulator (digikey.com and also other components) ensures accurate thermistor readings by minimising self-heating. A regulated high-current pulse is delivered to the heating element via a source built around an OP07 op-amp and an IRF5Y540CM MOSFET. The circuit’s performance and linearity were verified using simulation tools like LTspice [44]. Data acquisition relies on an Arduino Nano (techfun.sk and also other components) [45] interfaced with an ADS1256 24-bit analogue-to-digital converter [46] to precisely measure the voltage across the thermistors. Temperature values are calculated using the Steinhart–Hart equation, applying coefficients derived from the thermistor datasheet [43] and calibration procedures. A graphical user interface developed in Node-RED [47], running on a controlling notebook or potentially a Raspberry Pi for portability, manages the entire process. It initiates the heat pulse, logs temperature–time data, performs real-time Fast Fourier Transform (FFT) analysis for noise reduction, and calculates the thermal conductivity coefficient (*k*) based on the heat energy (*Q*), sample thickness (*d*), heater area (*S*), and the measured temperature difference (ΔT) and time lag (Δt) between the thermistor signal peaks, using the relation k=Q×d/(S×ΔT×Δt).

The obtained results demonstrated an increasing trend in thermal conductivity with higher AlN content: 0.21±0.01 W·$m^{-1}\cdot$$K^{-1}$ for the pure sample, 0.25±0.01 W·$m^{-1}\cdot$$K^{-1}$ for 10% AlN, 0.37±0.01 W·$m^{-1}\cdot$$K^{-1}$ for 20% AlN, and 0.53±0.01 W·$m^{-1}\cdot$$K^{-1}$ for 40% AlN. Each sample underwent more than ten measurements to ensure statistical validity. The heating pulse duration and power were optimised to maintain peak temperatures below 50 °C—the optimal operating range for the chosen thermistors [43]—although supplementary tests up to 100 °C confirmed the consistency of results. Variability in the pressure applied by the torque adapter was identified as the principal source of statistical error.

Validation procedures confirmed the setup’s accuracy and reliability. Comparative measurements against a professional TPS 2500 S (Hot Disk Instrument, USA) showed good agreement (e.g., 0.21±0.01 W·$m^{-1}\cdot$$K^{-1}$ in comparison to 0.2096±0.0002 W·$m^{-1}\cdot$$K^{-1}$ for pure PUR). Testing with a standard reference material (k=0.2 W·$m^{-1}\cdot$$K^{-1}$) verified the targeted 5% accuracy. The system’s robustness was established by its stability against ambient temperature changes within the 20 °C to 30 °C range, where no significant measurement variations were detected. The effectiveness of real-time data processing was confirmed by the negligible difference observed between online and offline analysis results.

### 3.4. Dielectric Permittivity Measurements

Dielectric characterisation was performed via frequency-domain dielectric spectroscopy using two instruments to cover a wide frequency range: IDAX 350 (Megger Sweden) (1 mHz to 10 kHz) and the 7600 Plus Precision LCR Meter (QuadTech, USA) (1 kHz to 1 MHz). This dual-instrument approach enabled a comprehensive analysis of dielectric relaxation, polarisation mechanisms, and interfacial effects in the composites.

The disk-shaped sample (80 mm diameter, 1.2 mm thickness) was placed in a temperature-controlled chamber (accuracy of ±0.5 °C) and measured over a temperature range from 20 °C to 120 °C. Standard protocols for dielectric spectroscopy were followed [48,49], with the real part of the dielectric permittivity ($\varepsilon_{r}$) and the loss tangent (tanδ) extracted from the complex permittivity spectra.

Electric measurements were performed using current and voltage techniques. The raw current and voltage data were digitised to enable a microprocessor to accurately compute the relevant electrical parameters. Each complete measurement cycle required approximately 0.5 s. For a full measurement, a cycle encompassed seven distinct individual measurements. These measurements include the following:Voltage Measurement ($V_{p}$): 0° and internal gain factor setting;Voltage Measurement ($V_{q}$): 90°;Reference Measurement: 0°;Gain factor > 1                  Gain factor = 1;Reference Measurement: 90°            Current Measurement: 0°;Current Measurement ($I_{p}$): 0°         Current Measurement ($I_{q}$): 90°;Current Measurement ($I_{q}$): 90°         Reference Measurement ($I_{p}$): 0°;Reference Measurement 0°                Reference Measurement: 90°.

In each measurement cycle, the following components were determined: $V_{p}$, $V_{q}$, $I_{p}$, $I_{q}$ (Figure 3a). Based on the given values, a diagram could be constructed, from which the phase angle ϕ and α between *I* (current) and *V* (voltage) could then be determined [50].

From measured values presented in the diagram, the microprocessor calculated the equivalent series resistance $R_{s}$ and the equivalent series reactance $X_{s}$ (Figure 3b) via the following equations:(10)$$R_{s}=\frac{V_{p}I_{p}+V_{q}I_{q}}{I_{p}^{2}+I_{q}^{2}},\;X_{s}=\frac{V_{q}I_{p}+V_{p}I_{q}}{I_{p}^{2}+I_{q}^{2}},$$ The calculated values determined the quality factor: $Q=tg\varphi =\frac{1}{D}=\frac{|X_{s}|}{R_{s}}$.

If the series reactance is negative, then our sample has capacitance properties; dominant C and secondary R parallel parameters can be calculated by the following formulas:(11)$$R_{p}=\left(1+Q^{2}\right)R_{s},\;C_{p}=\frac{1}{2\pi f\left(1+1/Q^{2}\right)\left|X_{s}\right|},$$

Based on the measured values of $C_{p}$ and $R_{p}$ from the LCR meter, tg δ and real permittivity can be calculated using the following formulas:(12)$$tan\;\delta =\frac{1}{2\pi fC_{p}R_{p}},\;\varepsilon_{r}=C_{p}/C_{0},$$
where $C_{0}$ is the air gap capacitance determined based on the sample thickness. In the case of IDAX, this instrument automatically calculates tanδ from the measured real and imaginary capacitance of the sample.

## 4. Simulation and Experimental Data

### 4.1. Percolation

To enable efficient energy transfer in the discussed composites, the particle concentration must be sufficient to reach the percolation threshold. Figure 4 presents a microtome slice of a sample with 20 wt% AlN, corresponding to a filler volume fraction of φ≈0.08, which is well below the percolation threshold. In Figure 4a, the original microscopic image is shown with a 100 μm resolution. In Figure 4b, points from the left image associated with AlN particles are labelled in red. It can be observed that individual chains of particles were already forming, which naturally tend to develop into longer chains at higher concentrations. Since a 3D representation of such conditions under natural settings is complex, the percolation cluster was simulated in a 3D probe cube.

The simulation was implemented with the following parameters and assumptions. The simulation box for the percolation system measures 150 μm on each side. Particles are assumed to be perfectly spherical with a diameter D = 10 μm. A connection between two particles is considered active if the distance between their centres is less than or equal to D+0.2D=1.2D (i.e., 12 μm), meaning their surfaces are within 2 μm of each other. This distance is justified because direct contact between particles may be hindered by microhydraulic and viscous effects of the liquid polyol between particles during processing. The influence of changes in the speed of energy transmission through phonon vibrations in the solid sample after manufacturing is considered negligible.

Percolation occurs when a cluster of connected particles spans the simulation box, connecting opposite faces in at least one direction (x, y, or z). The simulation results indicate that the percolation threshold is achieved at a volume fraction of approximately 0.22, which aligns with analytical predictions for random sphere packings adjusted for the connectivity criterion. As the particle concentration increases above the percolation threshold, the efficiency of the percolation network saturates in regions of high particle contact.

The probability (Figure 5) of achieving percolation exceeds 50% near a volume fraction of φ=0.28, suggesting that the upper analytical limit of 0.29 is of significant industrial interest for manufacturing thermally conductive composites. Three simulation cases are shown in Figure 6. The first one is for φ=0.08, which is well below the percolation threshold. Here, an absence of percolation clusters can be seen, which means that an insignificant increase in energy transmission through the polymer is due to a shorter distance between particles, where energy is less dissipated by the crystalline lattice of particles in comparison to molecular chains of the polymer.

The second one is for φ=0.22, which is within the percolation threshold. Here, it can be seen that percolating clusters occur. Here, in addition to high concentration, the percolation network ensures increased efficiency in energy transmission, since particles, in the first approximation, are in direct contact, ensuring highly efficient energy transmission. Finally, the last diagram shows that the concentration corresponds to a volume fraction of about 0.36, which is higher than the percolation threshold, making the percolation effect saturated and more efficient.

To study the influence of particle diameter on percolation, it was decreased up to 1 μm. For this case, the behaviour differs slightly. For D = 1 μm, the percolation threshold is φ=0.27. (Figure 6d). Here, the box size was set up 50 μm, and the distance between particles was about 0.1 D. Smaller particle sizes enable better matrix filling by the filler. However, achieving the percolation threshold requires higher particle concentrations, which automatically increases the overall filler content in the matrix. This leads to improved energy transmission efficiency due to the following:Reduced mean distance between particles;Better particle distribution;Higher overall concentration.

These factors suggest that smaller particle diameters are advantageous. Nevertheless, producing smaller particles requires greater industrial complexity and may increase manufacturing costs.

The simulation program was developed in Python, utilising the Random Sequential Addition algorithm to arrange particles and a Depth-First Search algorithm to identify clusters of connected particles. Each iteration involved incrementally increasing the number of particles and checking for possible connections between them.

### 4.2. Thermal Conductivity

To clarify the analysis, only the most relevant analytical models are discussed. For instance, the Ngo–Byon model is applicable only at low filler volume fractions and exhibits nonphysical behaviour at higher loadings, leading to its exclusion. Likewise, although the Effective Medium Theory [33] and Lewis–Nielsen models [34] have their merits, their applicability was limited in the context of this study. While the modified Hashin–Shtrikman (HS) model [37] shows potential for hybrid composites and will be considered in future work, it is not included in the present analysis. Notably, although the Wiener bounds yield curves similar to those of the HS model, the experimental data align more closely with HS predictions. Among the various analytical approaches, the model developed by Nan et al. [21] is considered the most appropriate because it accounts for filler particle shape and contact resistivity. The corresponding curve, shown in Figure 7, is described by the following expression:(13)$$k_{c}=k_{m}\frac{3+f\left[2\beta_{11}\left(1-L_{11}\right)+\beta_{33}\left(1-L_{33}\right)\right]}{3-f\left(2\beta_{11}L_{11}+\beta_{33}L_{33}\right)}$$
where(14)$$\beta_{11}=\frac{k_{11}^{c}-k_{m}}{k_{m}+L_{11}\left(k_{11}^{c}-k_{m}\right)},\;\beta_{33}=\frac{k_{33}^{c}-k_{m}}{k_{m}+L_{33}\left(k_{33}^{c}-k_{m}\right)},$$(15)$$k_{11}^{c}=\frac{k_{f}}{1+\gamma L_{11}k_{f}/k_{m}},\;k_{33}^{c}=\frac{k_{f}}{1+\gamma L_{33}k_{f}/k_{m}},$$(16)$$L_{33}=2-L_{11}=1-2\left[\frac{\rho^{2}}{2(\rho^{2}-1)}-\frac{\rho }{2{\left(\rho^{2}-1\right)}^{3/2}}\cosh^{-1}\rho \right].$$ Here, $k_{c}$ represents the isotropic effective thermal conductivity due to random filler dispersion, $k_{m}$ and $k_{f}$ denote the thermal conductivities of the matrix and filler, respectively, and $k_{\mathrm{ii}}^{c}$ (with i=1,3) represent the equivalent directional thermal conductivities. The parameter *f* is the filler volume fraction, and $\rho (=a_{3}/a_{1})$ is the aspect ratio (thickness to diameter) of the filler particles. The factor γ=α(1+2ρ), where $\alpha =a_{k}/a_{3}$, with $a_{k}=k_{m}/G_{k}$ and $G_{k}$ being the interfacial thermal conductance. In this analysis, particles are assumed to be perfectly spherical, resulting in shape factors $L_{11}=L_{33}=1/3$ and an interfacial thermal conductance of the order of $10^{7}$ W·$m^{-1}\cdot$$K^{-1}$.

The extended tunability of the Nan model, particularly its ability to account for interfacial thermal conductance between metal microparticles and the PUR matrix, enables significantly improved accuracy in predicting thermal conductivity (*k*) compared to classical percolation theory. As illustrated in Figure 8, the classical percolation model overestimates *k*, exhibiting a sharp increase at higher filler volume fractions within the percolation threshold. In contrast, the Nan model aligns closely with the experimental data (Table 1; Figure 7), which show a more gradual enhancement of *k*. This discrepancy arises because the percolation model assumes ideal thermal contact (infinite interfacial conductance) between particles and the matrix, whereas the Nan model explicitly incorporates finite interfacial thermal resistance, a critical factor in real composites. This resistance arises at the interfaces between the polymer matrix and filler particles, impeding heat flow due to poor phonon coupling. Thermal performance may benefit, for example, from surface functionalization. Although not directly applied in this study, this approach is a widely used strategy to mitigate Kapitza resistance by chemically modifying particle surfaces to enhance bonding with the surrounding matrix. Improved interfacial bonding facilitates more efficient phonon transmission, potentially increasing the composite’s thermal conductivity. In PUR/AlN/wBN composites, optimising particle–matrix interactions could similarly reduce Kapitza resistance, thereby enhancing the overall thermal performance for applications that require efficient heat dissipation.

The superior agreement of the Nan model is further demonstrated in Figure 7, where its predictions for *k* across varying volume fractions closely match the experimental trends. By integrating tunable parameters such as interfacial conductance, particle geometry (e.g., aspect ratio), and matrix–filler interactions, the Nan model captures the nonlinear behaviour of thermal transport in high-concentration composites. This capability is essential for designing battery enclosures, where balancing heat dissipation (governed by *k*) and electrical insulation (dependent on dielectric properties) is critical.

The observed higher thermal conductivity of wBN-loaded composites implies that higher thermal performance can be achieved with a lower volume fraction compared to AlN, thereby simplifying the mixing process. As could be seen from the predictive diagram in Figure 7, the industrially interesting value of thermal conductivity could be achieved starting from 0.5<φ<0.6, which represents an interesting problem of manufacturing and a question about the relevance of the application. These questions will be addressed in the next authors’ work.

### 4.3. Dielectric Measurement

The dielectric characterisation of the materials was performed by measuring their complex permittivity and dissipation factors using an IDAX and an RLC meter [5]. Measurements were evaluated in the temperature range of 25 °C to 120 °C. The results consistently reveal frequency-dependent dielectric parameters, a characteristic often observed in polar polymer matrices containing microparticles. This behaviour is indicative of Maxwell–Wagner–Sillars (MWS) polarisations, as well as the intermediate dipolar effect (IDE), and suggests the presence of DC conductivity near the electrodes [41]. MWS arises from the accumulation of charge carriers at interfaces between regions with different dielectric properties, PURs, and microfillers. The IDE relaxation is associated with the local motion of chain segments of the polar side groups around the C-C bond. While previous thermal analyses demonstrated that incorporating microparticles into PUR materials enhanced their thermal properties, the dielectric measurements present a more complex picture. Dielectric data for PUR composites containing AlN microparticles (0 wt% to 60 wt% by weight) and wBN microparticles (0 wt% to 40 wt% by weight) are presented in Figure 9 and Figure 10. A comparative analysis of both microfiller types was conducted at a fixed temperature of 60 °C to discern their impact on dielectric performance.

The development of dielectric parameters for various concentrations of AlN in PUR was similar, so here only the basic characteristic for 60% AlN in PUR is described (Figure 9a). At 25 °C, $\varepsilon_{r}$ slowly decreased from 8.5 to 4.3 to within the frequency range of 1 mHz to 1 MHz (Figure 9a). The real permittivity at the highest frequencies corresponds to the high-frequency limit, $\varepsilon_{\infty }$, as dipoles cannot respond sufficiently rapidly (Equation (Equation 5)). At temperatures 60 °C and higher, a DC conductivity near the electrodes became significant, resulting in a marked increase in $\varepsilon_{r}$ in the sub-hertz frequency range. With increasing temperature, this increase shifted to higher frequencies, whereas at frequencies above 10 Hz, there was minimal change in $\varepsilon_{r}$.

The effect of various relaxation processes can be better detected at the development of the loss tangent (Figure 9b). Notably, tanδ, representing energy dissipation, remained low across the examined frequency spectrum and temperatures, confirming minimal dielectric heating and energy losses—a key requirement for insulation applications. An increase in tanδ at very low frequencies is likely due to DC conductive losses, as expressed in Equations (Equation 3) and (Equation 5); however, these losses do not significantly impair the insulating properties of the composites. At 40 °C, the dielectric spectrum exhibited a single local maximum centred around 10 Hz, which was attributed to the IDE process. Upon increasing the temperature to 60 °C, a second local maximum emerged at approximately 10 mHz, indicative of MWS polarisation. Both of these characteristic relaxation peaks demonstrate a systematic shift towards higher frequencies with increasing temperature, as is consistent with thermally activated processes.

A critical temperature of 70 °C is often associated with the onset of thermal runaway in batteries, particularly lithium-ion chemistries. To ensure relevance to battery applications and facilitate direct comparisons, a temperature of 60 °C was used for evaluating the dielectric parameters of the PUR-AlN and PUR-wBN composite mixtures across various concentrations. From the influence of AlN on the real permittivity results (Figure 9c), concentrations below 10% caused its reduction due to a decrease mobility of PUR chains [49]. A similar decrease in permittivity was observed in the case of an admixture in the form of MgO, ZnO, and ${\mathrm{SiO}}_{2}$ nanoparticles, but only for small concentrations up to 1 wt% [41]. Since nanoparticles have a large active surface area, they can bind many polymer chains, which reduces their mobility and therefore the permittivity. The decrease in chain mobility was also confirmed by Dynamic Mechanical Analysis [5]. For concentrations of AlN higher than 20%, increases in real permittivity for frequencies over 1 Hz were measured due to the higher permittivity of the microparticles. In composites with higher filler concentrations, particularly near the percolation threshold, MWS polarisation became significant, enhancing $\varepsilon_{r}$ at middle frequencies. However, even at filler concentrations up to 50 wt% (approximately ϕ∼0.27 for AlN), the increase in $\varepsilon_{r}$ remained moderate, indicating that either the percolation network was not fully developed or that polarization effects were partially suppressed. In the case of the loss tangent in our measurements (Figure 9d), samples with 50 wt% and higher filler content showed only small differences and only at low frequencies, which indicates that even in the percolation regime, the filler density was not sufficient to significantly deteriorate the insulation properties.

Polymer chains binding to microparticles change their mobility, which in turn affects polarisation processes. From the data in Table 2, it can be seen that in the case of the IDE process ($f_{2}$), the movements of the side groups of the chains were primarily affected. With increasing concentrations of AlN, the heterogeneity of the mobility of the polymer side chains increases, meaning those in the vicinity of the microparticles exhibit reduced mobility compared to those located in the free volume. This effect is manifested by an increase in the coefficient $\alpha_{2}$. For coefficient $\beta_{2}$, which corresponds to the differing properties and structure of the material, we observed the most significant change between 5 and 20 wt%. This is a consequence of the increasing entanglement of polymer chains with the admixture. In the case of MSW polarisation ($f_{1}$), the influence of the local field on the mobility of the entire segment of polymer chains was lower, which corresponded to the small change observed in coefficients $\alpha_{1}$ and $\beta_{1}$. At concentrations 20 wt% and higher, the microparticles’ effect became more prominent, resulting in a progressive increase in real permittivity.

In the case of wBN addition, we observed a different effect on the dielectric properties. For 10 and 20 wt% additions, the real permittivity increased, but for 40 wt%, a significant increase was observed. In the case of tanδ, only a decrease was observed. This was caused by boron nitride, which in general is known for its excellent electrical insulation properties, low dielectric constant, and high dielectric breakdown strength.

It is clearly seen from Table 3 that the position of the local maxima corresponding to the IDE process ($f_{2}$) was significantly affected by the presence of wBN. With increasing concentrations of wBN, the heterogeneity of the mobility of the polymer side chains became more pronounced, like for AlN. In the case of MSW polarisation, the influence of the local field on the mobility of the entire segment of polymer chains was more important than for AlN. The value of $\alpha_{1}$ was almost two times as big. An interesting effect was the decrease in DC conductivity with increasing concentration, which corresponds very well with Figure 10c, where we observe a general decrease of real permittivity. For concentrations of 50 wt% and more of AlN, we observed a similar decrease in conductivity (Table 2).

Overall, the dielectric properties of the PUR-AlN and PUR-wBN composites indicate that excellent insulation characteristics were maintained even at high filler loadings, making these materials highly suitable for applications where both thermal management and electrical insulation are critical.

## 5. Conclusions

The thermodielectric properties measured for the developed composites underscore their potential as advanced materials for battery technology. This study has systematically investigated the thermodielectric properties of PUR composites reinforced with AlN and wBN microfillers, providing critical insights into their potential for thermally conductive materials with good insulating properties. Experimental measurements across a wide range of filler concentrations reveal that PUR-AlN and PUR-wBN microcomposites achieve moderate thermal conductivities ($k_{\mathrm{eff}}$ up to 0.6–0.7 W·$m^{-1}\cdot$$K^{-1}$ for AlN at φ=0.27) while maintaining excellent dielectric insulation, with low loss tangents (tanδ) even at high filler loadings. The Nan model, incorporating interfacial thermal resistance, accurately predicted the gradual increase in $k_{\mathrm{eff}}$, aligning closely with experimental data, whereas classical percolation theory overestimated the thermal conductivity due to its neglect of phonon scattering at filler-matrix interfaces. Our dielectric permittivity measurements demonstrate robust insulation properties, with minimal energy losses across a broad frequency range (1 mHz to 1 MHz) and range of temperatures (25 °C
to 120 °C
), making these composites suitable for both DC and AC electrical regimes in battery systems.

Three-dimensional percolation simulations reveal the transition from isolated filler contributions to a connected network, demonstrating a percolation threshold of $\varphi_{c}\approx 0.27$ for 1 μm particles. This value is slightly higher than for 10 μm particles ($\varphi_{c}\approx 0.22$), reflecting the greater filler density required for smaller particles to achieve connectivity.

## Figures and Tables

**Figure 1 sensors-25-04055-f001:**
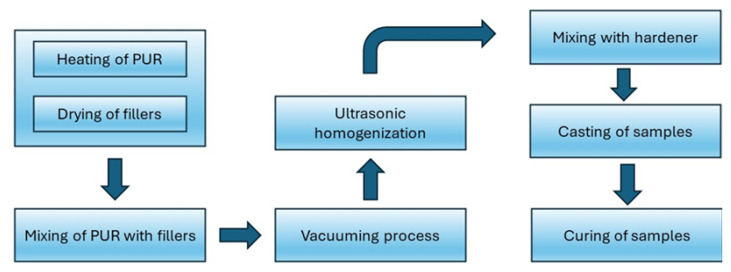
Procedure of sample preparation.

**Figure 2 sensors-25-04055-f002:**
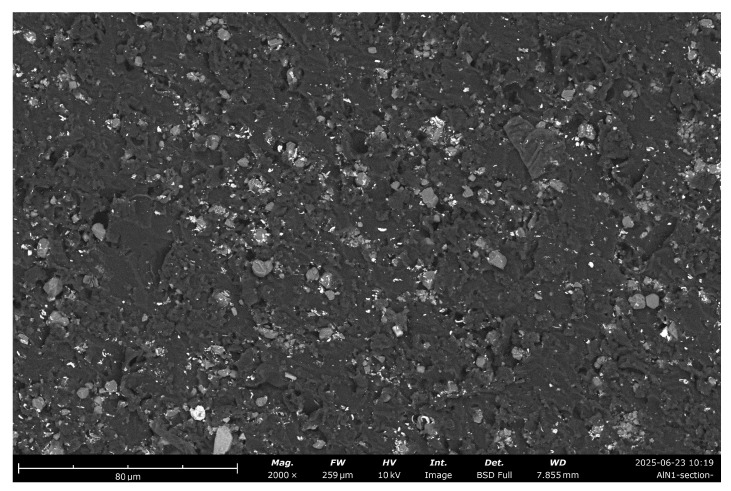
The SEM image with resolution 80 μm of sample PUR + 20 wt% of AlN corresponded to filler volume fraction φ=0.08.

**Figure 3 sensors-25-04055-f003:**
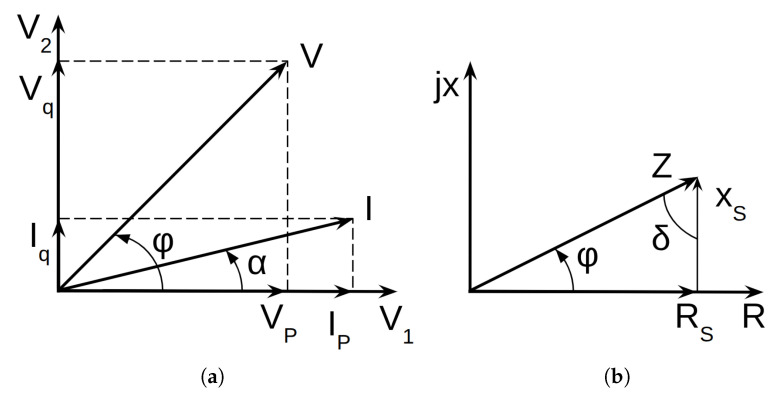
Diagrams (**a**,**b**) represent the results of LCR meter and explanation of $R_{s}$ and $X_{s}$ values.

**Figure 4 sensors-25-04055-f004:**
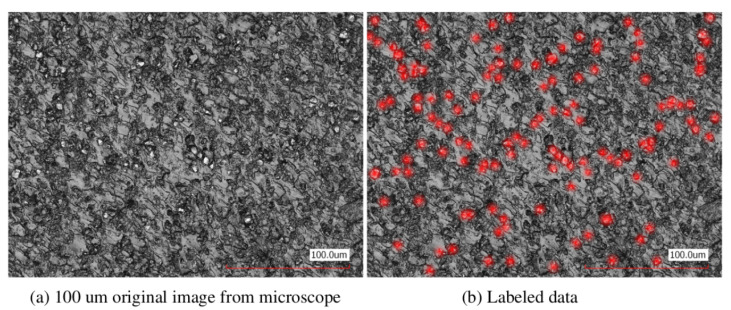
(**a**) The microscopic 100 μm image of a film-sliced PUR sample with 10 wt% of AlN corresponded to filler volume fraction φ=0.036. (**b**) The same image with red dots marking the AlN particles.

**Figure 5 sensors-25-04055-f005:**
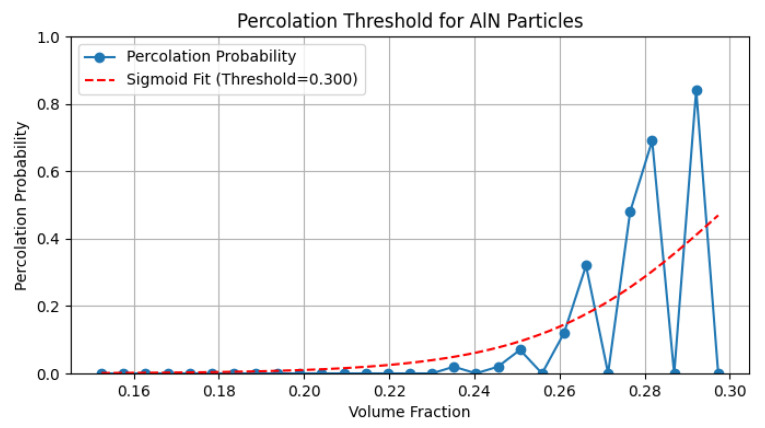
The diagram of the probability of percolation threshold on the example of AlN 10 μm particles.

**Figure 6 sensors-25-04055-f006:**
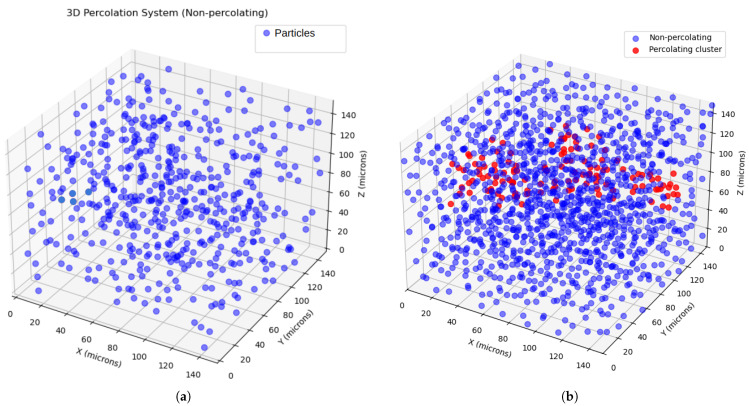
(**a**) shows 3D percolation model for volume fraction 0.08, below percolation threshold for particles 10 μm; (**b**) shows 3D percolation model for volume fraction 0.22, within percolation threshold for particles 10 μm; (**c**) shows 3D percolation model for volume fraction 0.36, beyond percolation threshold for particles 10 μm; (**d**) shows 3D percolation model for volume fraction 0.27, within percolation threshold for particles 1 μm; blue coloured particles are outside of percolation network, while red particles are within.

**Figure 7 sensors-25-04055-f007:**
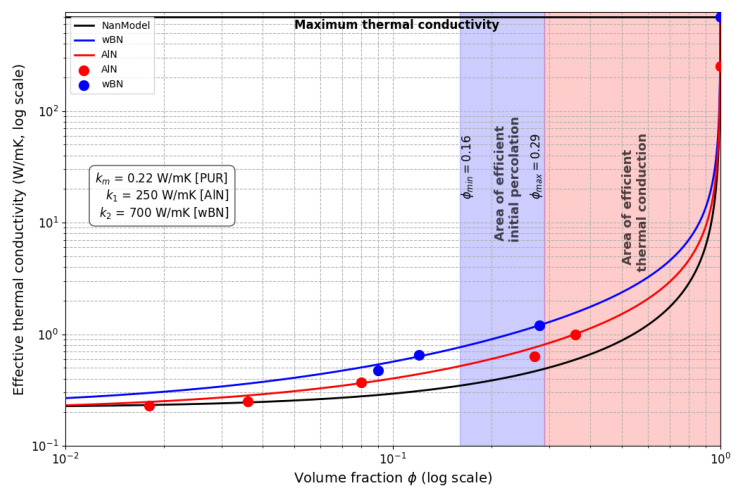
The diagram of dependence an efficient thermal conductivity $k_{\mathrm{eff}}$ on volume fraction ϕ. The black curve was generated from the Nan model for AlN for comparison with the red curve, predicted on the basis of experimental points (red) obtained for AlN. The blue curve was predicted on the basis of the Nan model for experimental points (blue) for wBN. Here, $k_{m}$, $k_{1}$, and $k_{2}$ denote the thermal conductivities of PUR, AlN, and wBN, respectively.

**Figure 8 sensors-25-04055-f008:**
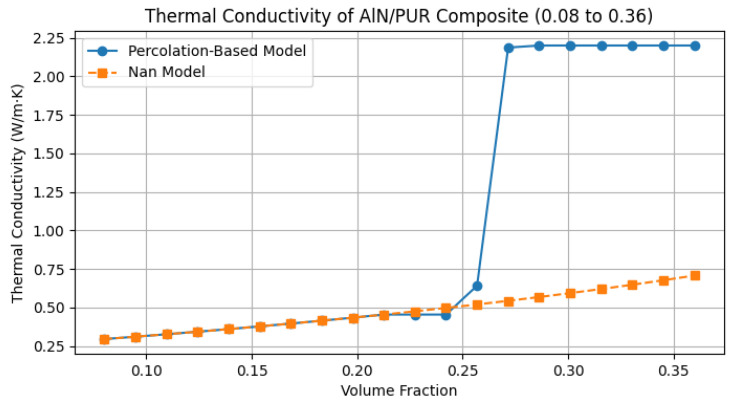
The comparison of thermal conductivity prediction from simulation of classical percolation threshold and Nan analytical model.

**Figure 9 sensors-25-04055-f009:**
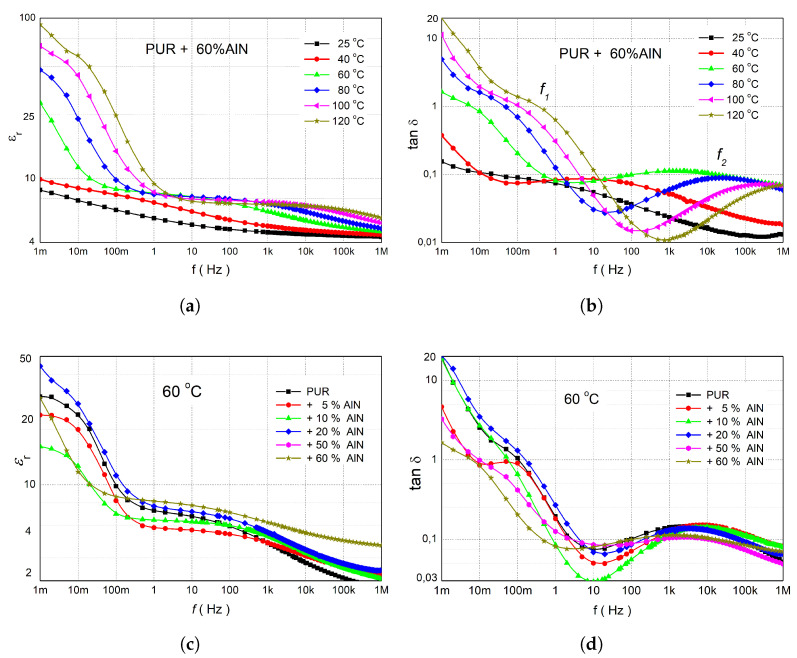
Figures (**a**,**b**) represent the behaviour of frequency-dependent real permittivity and loss tangent of PUR + 60% AlN, respectively, for a range of temperatures from 25 °C
to 120 °C. (**c**,**d**) represent real permittivity and loss tangent of PUR with various concentrations of AlN, respectively, only for the temperatures 60 °C.

**Figure 10 sensors-25-04055-f010:**
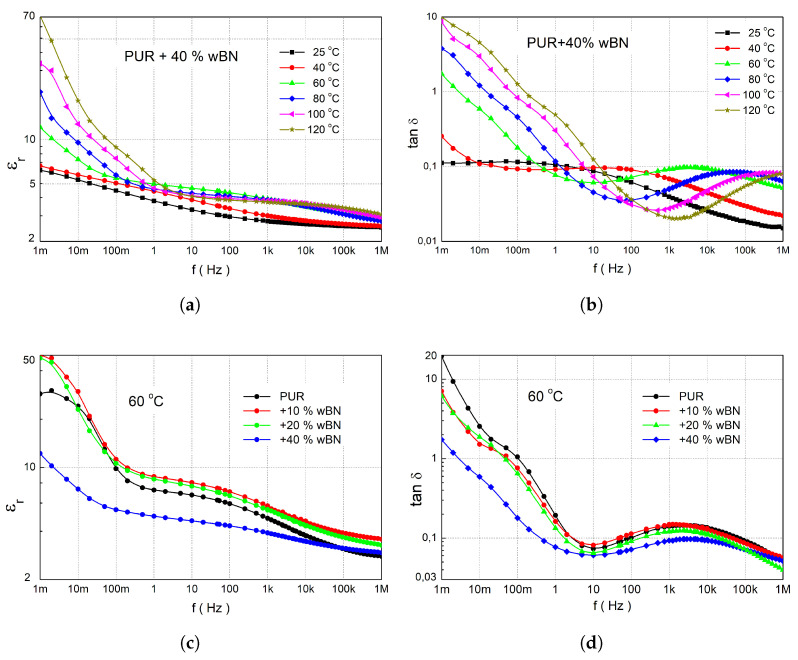
Figures (**a**,**b**) represent real permittivity and loss tangent of PUR + 40 % wBN, respectively, for a range of temperatures from 25 °C
to 120 °C. (**c**,**d**) represent real permittivity and loss tangent of PUR with various concentrations of wBN, respectively, only for the temperatures 60 °C.

**Table 1 sensors-25-04055-t001:** Measured thermal conductivities for samples with AlN and wBN fillers are presented. Values in cells marked with “-” indicate unmeasured samples. For higher concentrations, the mixing technique is complex and will be developed in the authors’ future work.

(%)	1	5	10	20	40	50	60
$\varphi_{\mathrm{AlN}}$	0.0035	0.018	0.036	0.08	0.2	0.27	0.36
$\varphi_{\mathrm{wBN}}$	0.004	0.02	0.04	0.12	0.28	0.35	0.45
$k_{\mathrm{AlN}}$ (W·$m^{-1}\cdot$$K^{-1}$)	0.21	0.23	0.25	0.37	0.53	0.63	0.72
$k_{\mathrm{wBN}}$ (W·$m^{-1}\cdot$$K^{-1}$)	0.21	-	0.35	0.65	-	-	-

**Table 2 sensors-25-04055-t002:** Parameters of the Havriliak–Negami, model for composites of various concentrations of AlN in PUR at temperature 60 °C, where $\varepsilon_{\infty }$ is the high frequency limit of the permittivity, σ ($10^{-12}$ S/m) is the DC conductivity, τ is the relaxation time in seconds, f = 1/(2πτ), and α and β are the shape parameters based on Equation (Equation 5). Index 1 corresponds to MWS polarisation and index 2 to IDE process.

(Par—wt %)	0	5	10	20	50	60
$\varepsilon_{\infty }$	2.51	2.43	2.47	2.98	3.51	4.47
$\Delta \varepsilon_{1}$	22.37	18.73	9.37	51.80	28.44	33.85
$\tau_{1}\left(s\right)$	6.42	5.50	13.03	35.14	28.25	102.81
$f_{1}$ (mHz)	25	29	12	5	6	2
$\alpha_{1}$	0.16	0.1	0	0.37	0.31	0.2
$\beta_{1}$	1	1	0.81	1	1	1
σ	31.7	6.11	16.6	51.2	5.49	1.95
$\Delta \varepsilon_{2}$	4.62	3.06	3.89	4.29	4.67	3.93
$\tau_{2}\left(ms\right)$	0.33	0.339	0.225	0.289	0.343	0.396
$f_{2}\left(Hz\right)$	483	470	708	551	464	402
$\alpha_{2}$	0.56	0.171	0.44	0.51	0.68	0.74
$\beta_{2}$	0.79	0.26	0.45	0.64	0.96	0.92

**Table 3 sensors-25-04055-t003:** Parameters of the Havriliak–Negami, model for composites of various concentrations of wBN in PUR at temperature 60 °C), where $\varepsilon_{\infty }$ is the high frequency limit of the permittivity, σ ($10^{-12}$ S/m) is the DC conductivity, τ is the relaxation time in seconds, f=1/(2πτ), and α and β are the shape parameters based on Equation (Equation 5).

(Par—wt %)	0	10	20	40
$\varepsilon_{\infty }$	2.51	2.95	2.88	2.62
$\Delta \varepsilon_{1}$	22.37	50.27	42.99	12.20
$\tau_{1}$ (s)	6.42	23.96	55.89	99.79
$f_{1}$ (mHz)	25	7	3	2
$\alpha_{1}$	0.16	0.33	0	0.39
$\beta_{1}$	1.00	1.00	0.58	1.00
σ	31.7	19.1	17.3	0.97
$\Delta \varepsilon_{2}$	4.62	4.93	4.79	2.30
$\tau_{2}$ (ms)	0.33	1.79	1.27	0.29
$f_{2}$ (Hz)	483	89	125	534
$\alpha_{2}$	0.56	0.25	0.43	0.63
$\beta_{2}$	0.79	0.29	0.44	0.71

## Data Availability

Data are contained within the article.

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
