# Peer review of "Thermodielectric Properties of Polyurethane Composites with Aluminium Nitride and Wurtzite Boron Nitride Microfillers: Analysis Below and near Percolation Threshold"

_sensors, 2025, doi:10.3390/s25134055_

Round 1
Reviewer 1 Report
Comments and Suggestions for Authors
The manuscript „Thermodielectric Properties of Polyurethane Composites with AlN and wBN Microfillers: Analysis Below and Near Percolation Threshold” deals with the preparation of polyurethane composites reinforced with aluminium nitride and wurtzite boron nitride in different mass percentages and investigation of their thermodielectric properties for potential application as advanced materials for battery technology. Still, this manuscript needs some corrections and clarifications before it can be considered for possible acceptance and publishing.
The comments for this opinion are listed below:
- Avoid using abbreviations that are not commonly used, not well known such as AlN wBN in the title of the manuscript.
- I propose that Section 2 can be moved to Supporting information, for better observation of the main results of this paper.
- In Section 3.1 should be emphasize which polyurethane (which isocyanate, polyol, solvents, possible chain extender, catalyst, etc,) was used for preparation of these composites.
- How the authors confirmed the existence of phase separation, between the segments, which has a major impact on the achieved final properties of the PUR composites, as stated in Inroduction Section? I suggest AFM analysis for confirmation, since the Figure 3 is not sufficient for this confirmation.
- Whether the authors prepared in this paper microcomposites or nanocomposites (line 196)? Nanofillers or microfillers? This should be the same throughout the entire manuscript. Moreover, due to the high loadings of the fillers inside the polyurethane matrix, which can have a major impact on the inhomogeneous distribution of added fillers, their dispersion should be verified by some morphological analyses such as TEM or SEM analysis.
Author Response
Comments 1: Avoid using abbreviations that are not commonly used, not well known such as AlN wBN in the title of the manuscript.
Response 1: It’s done by replacing with full names
Comments 2: I propose that Section 2 can be moved to Supporting information, for better observation of the main results of this paper.
Response 2: Thanks for the suggestion. Section 2 defines the necessary Theory for analyzing the measured results, without which it would be more complicated to explain the measured dependencies. Based on the classical sequence for research articles: Introduction (Part 1) - Theory (Part 2) - Experiments -... etc. I would keep the given sequence for this article as well.
Comments 3: In Section 3.1 should be emphasize which polyurethane (which isocyanate, polyol, solvents, possible chain extender, catalyst, etc,) was used for preparation of these composites.
Response 3: It’s done by clarifying “polyol”
Comments 4: How the authors confirmed the existence of phase separation, between the segments, which has a major impact on the achieved final properties of the PUR composites, as stated in Inroduction Section? I suggest AFM analysis for confirmation, since the Figure 3 is not sufficient for this confirmation.
Response 4: AFM analysis is a good method for characterizing surfaces. However, in this case, the given method is not suitable, since we focus on changing the bulk properties of polyurethane. The surface in Fig. 3 was obtained by cutting through the sample and presents a uniform distribution of particles in the material. We made AFM analysis on the area of 20 x 20 um2, where it is possible to observe the roughness of the surface after cutting and the connection of microparticles and polyurethane.
Comments 5: Whether the authors prepared in this paper microcomposites or nanocomposites (line 196)? Nanofillers or microfillers? This should be the same throughout the entire manuscript. Moreover, due to the high loadings of the fillers inside the polyurethane matrix, which can have a major impact on the inhomogeneous distribution of added fillers, their dispersion should be verified by some morphological analyses such as TEM or SEM analysis.
Response 5:
It’s done by removing prefix in words like “particle“ and replacing micro everywhere we mention fillers or composites.
We appreciate your suggestion regarding TEM and SEM analysis. Unfortunately, we do not currently have access to the necessary equipment to perform a TEM analysis of our samples. We have explored options, but are currently only able to provide SEM characterization. SEM analysis of the sample surface was added to the article, as Figure 2. At this time, we were only able to perform SEM analysis on one sample that was prepared for the given analysis. Samples with higher concentrations were prepared only for thermal and dielectric measurements.

Reviewer 2 Report
Comments and Suggestions for Authors
First of all, thank you for submitting your manuscript sensors-3698815 with title:
(Thermodielectric Properties of Polyurethane Composites with AlN and wBN Microfillers:
Analysis Below and Near Percolation Threshold) to the journal Sensors. I read your article
perfectly and this is my comment on improving it.
Major revision:
1. Please move this part of abstract to introduction part (Polymer composites
reinforced with aluminium nitride (AlN) and wurtzite boron nitride (wBN)
microparticles exhibit strong potential for industrial applications, particularlyin the
battery sector, due to their favourable thermodielectric properties.)
2. In line 118 we need perfect reference for this clame “Techniques such as surface
functionalisation (e.g., silane coupling agents) can potentially enhance Gk to 5 × 107
W/m2K, which in turn may significantly improve keff.”
3. The content from lines 231 to 237 must be moved to the Introduction section. Because
we need better background context more suitable for that part of the manuscript.
Also, in part of the transient heat pulse methodology put a strong scientific reference
to support your approach, like equations that it possible put in supporting information
or add in theory section (line 88).
Furthermore, it is necessary to include detailed information about the design and the
calibration method used for developing your custom-built device for thermal
conductivity measurement. Because look likes one transient approach used for
calculating thermal conductivity. Therefore, we need clearly explain how heat flux is
measured during the process.
To the best of my knowledge, NTC thermistors cannot directly measure heat flux.
This device is typically suitable for use in steady-state heat flow setups not for
transient measurements of this nature.
Also, for clarify moor Please add additional details such as:
- A schematic or photo of your setup
- The geometry and dimensions of the samples measured
- The reference standard material used for validation (particularly for low thermal
conductivity) - The protocol used to calculate the measurement uncertainty of your setup
We need these details for scientific validity and reproducibility of your
measurements.
4. Please clarify why some thermal conductivity data points in Table 1 were not
measured or yielded negligible values (After Line 282 ). Also, to describe more
fundamental parts like chain mobility suppression describe more detail of the
dielectric permittivity and loss tangent trends for example reduction at low AlN
concentrations.
5. While the Nan model is well-justified, we need comparison with other models such
as Lewis-Nielsen, Hashin-Shtrikman, therefore it useful to support the theoretical
framework. Also, the dielectric results could be contextualized with prior work on
similar composites such as PUR-ZnO or PUR-SiO2, to highlight novelty of article.
6. It is recommended discuss scenario about limits thermal conductivity related to
interfacial resistance (Kapitza) like surface functionalization. Also, the trade-off
between filler loading and processability (e.g., viscosity increase at high φ) should
be notified for industrial scalability.
7. Figures 6 and 7 thermal conductivity models, are referenced but not included in the
main text. Please check this issue. Also, the Havriliak-Negami parameters in Tables
2–3 are highly technical please add a brief interpretation of their physical
significance.
Minor revision
8. In the abstract could explicitly state the percolation thresholds identified Like φ ≈
… you can show the margin.
All of the above, this manuscript after Major Revisions would be suitable for publication
in the journal Sensors.

Author Response
Comments 1: Please move this part of abstract to introduction part (Polymer composites reinforced with aluminium nitride (AlN) and wurtzite boron nitride (wBN) microparticles exhibit strong potential for industrial applications, particularlyin the battery sector, due to their favourable thermodielectric properties.)
Response 1: It’s done
Comments 2: In line 118 we need perfect reference for this clame “Techniques such as surface functionalisation (e.g., silane coupling agents) can potentially enhance Gk to 5 × 10^7 W/m2K, which in turn may significantly improve keff.”
Response 2: It’s done by adding corresponding reference right in that sentence
Comments 3: The content from lines 231 to 237 must be moved to the Introduction section. Because we need better background context more suitable for that part of the manuscript.
Response : It’s done by removing that text in the end of the introduction.
Comments 3: Also, in part of the transient heat pulse methodology put a strong scientific reference to support your approach, like equations that it possible put in supporting information or add in theory section (line 88).
Furthermore, it is necessary to include detailed information about the design and the calibration method used for developing your custom-built device for thermal conductivity measurement. Because look likes one transient approach used for calculating thermal conductivity. Therefore, we need clearly explain how heat flux is measured during the process.
To the best of my knowledge, NTC thermistors cannot directly measure heat flux. This device is typically suitable for use in steady-state heat flow setups not for transient measurements of this nature.
Also, for clarify moor Please add additional details such as:
- A schematic or photo of your setup
- The geometry and dimensions of the samples measured
- The reference standard material used for validation (particularly for low thermal
conductivity) - The protocol used to calculate the measurement uncertainty of your setup
We need these details for scientific validity and reproducibility of your
measurements.
Response 3:
We appreciate the reviewer's diligent attention to detail regarding the thermal conductivity measurement methodology. We would like to politely direct the reviewer to our previously published article, "Novel, Cost Effective, and Reliable Method for Thermal Conductivity Measurement", available at https://doi.org/10.3390/s24227269, where all the requested information is comprehensively addressed.
Here's a breakdown of where the information can be found:
- Detailed information about the design and calibration method for the custom-built device:
- Design: Section 2, "Setup," provides an overview of the experimental setup , including schematic illustrations. Specific details on the low-current source for thermistors are in Section 2.2 , and the high-current source for the heating element is in Section 2.3. The data acquisition system using Arduino Nano and ADS1256 is described in Section 2.4.
- Calibration Method: Section 2.5, "Steinhart-Hart Equation and Calibration," thoroughly explains the calibration of the NTC thermistors using the Steinhart-Hart equation and a fitting procedure from the datasheet. The setup was also validated using a standard reference material and through comparison with a professional instrument, the TPS 2500 S.
- How heat flux is measured during the process (transient approach):
- The article explicitly states that the thermal conductivity coefficient is calculated "from the heat generated, derived from the power flowing through the heating element, the sample dimensions, and the observed temperature difference and corresponding time difference peak spectra".
- Equation (1) k=Q*d/(S*ΔT*Δt) is provided, defining the calculation where Q is the power from the heating element and S is its area, essentially defining the heat flux density involved in the calculation. The Arduino program measures the voltage drop to obtain the power (Q).
- The method described is a transient approach, with a "heat pulse generated by a plane source" and thermistors measuring temperature at "pre-configured time intervals".
- Concern about NTC thermistors suitability for direct heat flux measurement and transient measurements:
- The NTC thermistors in this setup are used to "monitor the temperature on both sides of the sample", not to directly measure heat flux.
- The article details the use of NTC thermistors for temperature readings in their transient measurement setup. It also addresses the critical aspect of minimizing self-heating effects by using a low-current source.
- The paper confirms the device's reliability and stability for measurements in a relevant temperature range, mentioning that the "heating pulse was optimized to remain below 50 degrees Celsius to ensure the thermistor operates within its optimal temperature range, as it provides the highest accuracy up to this temperature". Additionally, tests up to 100 degrees Celsius showed consistent results, demonstrating the setup's robustness. Noise reduction and data smoothing techniques are also employed to ensure reliable temperature measurements in the transient process.
The size of the samples used was 50×50×3 mm.
Comments 4: Please clarify why some thermal conductivity data points in Table 1 were not measured or yielded negligible values (After Line 282 ). Also, to describe more fundamental parts like chain mobility suppression describe more detail of the dielectric permittivity and loss tangent trends for example reduction at low AlN concentrations.
Response 4:
Part 1 (data points): It’s done by adding to the figure’s capture text correspondent clarification
Part 2 (chain mobility + permittivity): Microparticles dispersed within a polymer matrix exhibit interactions with the polymer chains, leading to their immobilization. This immobilization results in a significant reduction in the mobility of the bound polymer chains, which subsequently manifests as a decrease in permittivity. Given the direct correlation between permittivity and the dynamics of polymeric polar chains, the reduced mobility of these chains directly contributes to the observed decrease in permittivity within the composite material. The reduction of loss tangent is associated with decrease of the micro-Brownian motion of segments in polymeric chains and the localized movement of molecules
Comments 5: While the Nan model is well-justified, we need comparison with other models such as Lewis-Nielsen, Hashin-Shtrikman, therefore it useful to support the theoretical framework. Also, the dielectric results could be contextualized with prior work on similar composites such as PUR-ZnO or PUR-SiO2, to highlight novelty of article.
Response 5:
Part 1 (models):The required text was added on page 5
Part 2 (similar composites): A similar decrease in permittivity was observed in the case of an admixture in the form of ZnO, SiO2 nanoparticles, but only for small concentrations up to 1%. Since nanoparticles have a large active surface area, they can bind many polymer chains, which reduces their mobility and therefore the permittivity. The decrease in chain mobility was also confirmed by DMA measurements of H1 spectra.
Comments 6: It is recommended discuss scenario about limits thermal conductivity related to interfacial resistance (Kapitza) like surface functionalization. Also, the trade-off between filler loading and processability (e.g., viscosity increase at high φ) should be notified for industrial scalability.
Response : Thank you for these comments. The required text about interfacial resistance was added on page 14.
With the increase in fillers, the viscosity of the resulting composite also increases. For this reason, we used two types of mixing in the mixing process to ensure sufficient homogeneity of the resulting distribution of microparticles. For the maximum concentration used in this article, this technological procedure was sufficient and does not require further improvements. In the case of higher concentration, however, it is necessary to add another technological step, right at the beginning. Before adding the microparticles to the polyurethane, it is necessary to mix them with a suitable liquid dispersant, which ensures the formation of a liquid paste. This paste is then mixed in the polyurethane without any major problems. However, the selection of a suitable dispersant, which ultimately does not affect the resulting properties of the composite, is the subject of further research.
Comments 7: Figures 6 and 7 thermal conductivity models, are referenced but not included in the main text. Please check this issue. Also, the Havriliak-Negami parameters in Tables 2–3 are highly technical please add a brief interpretation of their physical significance.
Response 7:
Description of Figure 7 (previous 6) is on page 14, line 441 and for Figure 8 (previous 7) on same page and 15.
The characterization of Havriliak-Negami parameters was added after the Equation (5) on the page 4. An explanation of the impact of the given parameters and their values presented in Tables 2 and 3 has been added to the article on page 17 and following.
Comments 8: In the abstract could explicitly state the percolation thresholds identified Like φ ≈… you can show the margin.
Response : The abstract was updated.
Reviewer 3 Report
Comments and Suggestions for Authors
The authors presents detailed examination of thermodielectric properties of PUR composites with two conductive fillers. In my opinion the work is worth to publish, although there are some questions that I would like to be explained.
The characteristics of the obtained composite materials, in particular the morphology, should be presented. This is important in the context of dielectric studies as well as composites with a very high filler content.
From the point of view of the utility of the presented materials. How the addition of the fillers affects the mechanical properties of the PU matrix, especially in the case of their biggest contents?
Why for the conductivity measurements authors used different concentrations of AlN filler?
I propose to introduce the basic properties of the used materials instead of cite a link to the manufacturer's website. The manufacturer's offer is subject to change as is the website address.
I also have some suggestions regarding editing:
- standardize presentation of numerical data with units (with or without gap),
- standardize accuracy of numerical data representation (significant numbers).
Author Response
Comments 1: The characteristics of the obtained composite materials, in particular morphology, should be presented. This is important in the context of dielectric studies as well as composites with a very high filler content.
Response 1: Thanks for the valuable feedback. We've incorporated SEM analysis for one sample type, which we acknowledge is a starting point. Our primary focus in this paper remained on the thermal and dielectric properties. Moving forward, our research will broaden to include mechanical and other characteristics of the composite materials, with careful attention to sample preparation for those specific measurements. We agree entirely that morphology becomes particularly crucial for composites with very high filler content.
Comments 2: From the point of view of the utility of the presented materials. How the addition of the fillers affects the mechanical properties of the PU matrix, especially in the case of their biggest contents?
Response 2: Thank you for your question. This study primarily focuses on the study of thermal and dielectric properties of polyurethane with AlN and wB additives. Theory and experimental measurements clearly show that both additives increase thermal conductivity and influence the complex permittivity and loss tangent. The given dielectric parameters depend greatly on the concentration of the additive. In the next study, we will focus on the complex properties of polymer micro composites, such as mechanical properties, fire resistance (battery encapsulation material) and other technical parameters.
Comments 3: Why for the conductivity measurements authors used different concentrations of AlN filler?
Response 3: We did not precisely grasp the main point of this question. Already from the article’s name it might be seen that different concentrations denote different levels of fillers content corresponding to closeness or presence within percolation level. The alteration of such a fundamental feature as thermal conductivity depending on different concentration (also volume fraction) is the gist. It must be noted that for wBN we also used different concentrations (see table 1) to show similar dynamics.
Comments 4: I propose to introduce the basic properties of the used materials instead of cite a link to the manufacturer's website. The manufacturer's offer is subject to change as is the website address.
Response 4: The basic parameters of VUKOL N22 MAGNA BLUE was added to the article, in Section 3.1
Comments 5: I also have some suggestions regarding editing:- standardize presentation of numerical data with units (with or without gap),
Response : Thank you for this comment, throughout the article we have standardized the presentation of numerical data using units.
Round 2
Reviewer 1 Report
Comments and Suggestions for Authors
The authors have made an effort to improve and clarify this manuscript in the best possible way. Therefore, I suggest that this manuscript could be considered for acceptance in this journal.
Author Response
Thank you.
Reviewer 2 Report
Comments and Suggestions for Authors
First of all, thank you for submitting your revised version of manuscript sensors-3698815 with updated title:
Thermodielectric Properties of Polyurethane Composites with Aluminium Nitride and Wurtzite Boron Nitride Microfillers: Analysis Below and Near Percolation Threshold
The authors answered all my questions and requests for new characterization. They can convince me mostly of my question, but about comment 3, I do not strongly agree with your method you measuring thermal conductivity, and also your article references. However, I still have some reservations about their method for measuring thermal conductivity, as well as the references cited in the article. Nevertheless, at this stage, I consider it a valuable step toward future work.
Additionally, I recommend that the last paragraph of the introduction focuses just on the novelty of your article, and that any discussion of missing work be placed before it.
All of the above, with minor revisions in part of the introduction, I recommend this article to the editor for publication in the journal Sensors.
Best Regards

Author Response
Comment 1: The authors answered all my questions and requests for new characterization. They can convince me mostly of my question, but about comment 3, I do not strongly agree with your method you measuring thermal conductivity, and also your article references. However, I still have some reservations about their method for measuring thermal conductivity, as well as the references cited in the article. Nevertheless, at this stage, I consider it a valuable step toward future work.
Response 1:We express our gratitude to the referee for their valuable and insightful comments.
Regarding the reviewer's recommendation, the original references 48 and 49 have been removed from the article.
Indeed, the thermal measurement method presented is novel and in the future we will focus on confirming it with further measurements. However, we note that a comprehensive analysis of this method was provided in our previous publication, which specifically detailed the thermal measurement techniques. We wish to reiterate that almost all measurements were validated using the TPS2500S, a widely recognized, high-performance scientific measurement system.
This article focuses mainly on investigating the thermodielectric properties of specific fillers and we used an accepted thermal conductivity measurement technique that has been realized and is used in our department.
We would be delighted to further explore the topic of thermal measurement techniques in our future work.
Comment 2: Additionally, I recommend that the last paragraph of the introduction focuses just on the novelty of your article, and that any discussion of missing work be placed before it.
Response 2: Based on the recommendation, we are adding a final paragraph of the introduction that focuses solely on the novelty of your article.
This study explores the thermal and dielectric properties of composites incorporating microparticles of AlN and wBN, focusing on their behaviour in the pre-percolation and efficient percolation regimes. The observed trends exhibit a strong correlation with analytical models of thermal conductivity as a function of filler volume fraction, providing critical insights into the material’s performance. These findings lay a solid foundation for the development of large-scale industrial applications, particularly for high-performance thermally conductive materials in next-generation battery systems. Furthermore, the results offer a reliable platform for future experiments exploring the hybridisation of AlN and wBN fillers to investigate potential synergistic enhancements.
Reviewer 3 Report
Comments and Suggestions for Authors
The work meets the journal’s scope. The authors answered all the questions requested, from my perspective, the article can be published.
Author Response
Thank you.